# Resting-State Functional Connectivity following Phonological Component Analysis: The Combined Action of Phonology and Visual Orthographic Cues

**DOI:** 10.3390/brainsci11111458

**Published:** 2021-11-02

**Authors:** Michèle Masson-Trottier, Anna Sontheimer, Edith Durand, Ana Inés Ansaldo

**Affiliations:** 1Centre de Recherche de l’Institut Universitaire de Gériatrie de Montréal, Montréal, QC H3W 1W5, Canada; michelemt8@gmail.com; 2Faculté de Médecine, Université de Montréal, Montréal, QC H3T 1J4, Canada; 3Centre National de la Recherche Scientifique, Institut National Polytechnique-Clermont, Institut Pascal, Université Clermont Auvergne, F-63000 Clermont-Ferrand, France; asontheimer@chu-clermontferrand.fr; 4Centre Hospitalier Universitaire de Clermont-Ferrand, F-63000 Clermont-Ferrand, France; 5U.F.R. Lettres, Cultures et Sciences Humaines, Université Clermont Auvergne, F-63000 Clermont-Ferrand, France; edith.durand@uca.fr

**Keywords:** aphasia, anomia, therapy, resting-state fMRI, French

## Abstract

Anomia is the most frequent and pervasive symptom for people with aphasia (PWA). Phonological component analysis (PCA) is a therapy incorporating phonological cues to treat anomia. Investigations of neural correlates supporting improvements following PCA remain scarce. Resting-state functional connectivity (rsFC) as a marker of therapy-induced neuroplasticity has been reported by our team. The present study explores the efficacy of PCA in French and associated therapy-induced neuroplasticity using whole-brain rsFC analysis. Ten PWA participated in a pre-/post-PCA fMRI study with cognitive linguistic assessments. PCA was delivered in French following the standard procedure. PCA led to significant improvement with trained and untrained items. PCA also led to changes in rsFC between distributed ROIs in the semantic network, visual network, and sub-cortical areas. Changes in rsFC can be interpreted within the frame of the visual and phonological nature of PCA. Behavioral and rsFC data changes associated with PCA in French highlight its efficacy and point to the importance of phonological and orthographic cues to consolidate the word-retrieval strategy, contributing to generalization to untrained words.

## 1. Introduction

Aphasia is an acquired language disorder following brain injury occurring in over a third of hospitalized stroke patients; an estimated 165,000 to 380,000 Canadians are affected with chronic aphasia [1]. Typically, aphasia results from damage in the left fronto-temporo-parietal brain regions. Among the range of symptoms, anomia—a difficulty to find words—is the most common, persistent, and debilitating symptom. Aphasia leads to reduced quality of life [2] and increased risk of depression [3]. Current knowledge shows that speech-language therapy effectively improves communication abilities (see Brady et al. [4] for a review), with outcome variability following treatment observed between persons with aphasia. Studies show that factors such as initial severity or size of lesion, known as an intrinsic variable, may impact recovery following stroke [5,6,7]. However, results have been inconsistent, and the recovery mechanisms remain poorly understood [8]. To better understand this variability in recovery, researchers have turned to neuroimaging and aim to uncover the neurophysiological changes induced by extrinsic variables such as therapy.

It is now well accepted that going beyond investigating the cerebral areas damaged by the stroke is necessary to comprehend language deficits and recovery in persons with aphasia following stroke. Language functions, namely word naming, are sustained by a dual-stream network, mainly lateralized in the left hemisphere (LH), including a ventral stream mapping sound to meaning (lexical representations) involving the superior temporal gyrus (STG), middle temporal gyrus (MTG), and inferior temporal gyrus (ITG) and a dorsal stream mapping sound to articulatory-based representations involving the STG, inferior frontal gyrus (IFG), middle frontal gyrus (MFG), angular gyrus (AG), supramarginal gyrus (SMG), and the supplementary motor area (SMA) [9,10,11,12,13]. These areas work as a distributed and complex network [11], along with domain-general networks [8]. In the event of a stroke, damage to the integrity of the dual-stream language network causes language difficulties such as anomia. However, after the stroke, the brain has the capacity to reorganize, namely through functional processes, following principles facilitating neuroplasticity such as specificity, salience, repetition and intensity, generalization, and complexity [7]. Early on, the brain will recruit right hemisphere (RH) homolog areas to execute the language tasks. This process has been both described as successful compensation and as maladaptive—leading to less recovery. In the chronic phase, it remains unknown if the RH recruitment is beneficial or not [8,14,15,16].

Speech-language therapies developed to improve naming abilities in people with aphasia are numerous. Therapies aiming to reduce anomia are generally based on semantic strategies, such as Semantic Feature Analysis [17,18], or phonological strategies, such as Phonological Component Analysis PCA [19,20,21,22,23]. In phonological therapies, the aim is to facilitate lexical retrieval by increasing the activation of the phonological representation of words [24]. The well-known PCA treatment protocol, developed by Leonard et al. [20], is well described and allows for reproducibility of results. Previous behavioral studies with PCA have established the treatment’s efficacy for trained items [19,20,23,25,26,27,28] and untrained items [20,23,25]. Neuroimaging studies have also demonstrated that the PCA protocol induces neuroplasticity in persons with aphasia [22,29,30]. However, PCA has never been studied in French, and the effect of PCA on whole-brain functional connectivity remains unknown.

Functional connectivity (FC) is a recent tool available to researchers to investigate the language network’s functional reorganization and, more specifically, the therapy-induced neuroplasticity. Traditionally, to study language networks, research paradigms employ task-based fMRI. However, performance during scan has been repeatedly mentioned as an important confounding factor with persons with aphasia [31,32]. More recently, resting-state functional magnetic resonance imaging (rs-fMRI) has been used to study language networks in persons with aphasia (see Klingbeil et al. [33] for review) and, more specifically, to explore therapy-induced neuroplasticity [34,35,36,37,38]. rs-fMRI has the advantage of creating a less stressful acquisition for the person with aphasia, being more easily reproduced in the clinical setting and not requiring the person with aphasia to accomplish a task during the acquisition [33,39]. To the best of our knowledge, only three previous studies report neuroimaging data following PCA therapy [20,27,36]. Marcotte et al. [22] compared FC changes after intensive PCA versus standard protocol administered to two participants. The improvements after intensive PCA were associated with decreased activations (right posterior cingulate gyrus, left precentral gyrus, left MFG) conjointly with increased activations (right caudate nucleus, left MFG), whereas the standard PCA protocol did not yield significant naming improvements. Rochon et al. [29] reported behavioral improvements associated with greater LH than RH processing and greater perilesional activity. From a segregative perspective, these findings are informative regarding activation therapy-induced neuroplasticity. However, a more integrative perspective, that considers that the language is implemented in widely distributed and functionally integrated large-scale networks and allows identifying post-therapy changes and network reorganization, is lacking [31]. To the best of our knowledge, the only study on resting-state functional connectivity (rsFC) changes following PCA is reported by van Hees et al. [38] using an amplitude of low-frequency fluctuations (ALFF) method. The results showed that ALFF values in the right and left MTG, the right IFG, and the left SMG are significantly correlated with PCA improvement. The ALFF method informs of local properties of the BOLD signal, but the neurophysiological basis of this measure remains poorly understood [31].

This study aims to use this well-established specific naming therapy, namely, PCA delivered in French (Fr-PCA), to better understand the brain connectivity changes associated with therapy-induced recovery. Specifically, we investigate the rsFC modifications after Fr-PCA. In this exploratory study, a whole-brain region of interest (ROI)-to-ROI analysis is performed to identify rsFC changes within and beyond the well-known language network. It is hypothesized that participants will improve in naming following Fr-PCA therapy and that the underlying processes for therapy will influence the rsFC. We investigate functional connections between separate brain regions (ROI-to-ROI). It is expected that following therapy, increased rsFC between ROIs within the language network in the dorsal stream, involved in phonological processing, and ROIs employed for semantic processing reflecting the work with words and pictures associated with meaning will be measured.

## 2. Materials and Methods

### 2.1. Participants

Ten participants with chronic aphasia (3 women, mean age = 68.9 ± 10.2) following a single left hemisphere ischemic stroke recruited through local patient association by referral following discharge from rehabilitation centers in the area and following self-referral are included in this study. Aphasia severity and typology are determined by an experienced speech-language pathologist (SLP; M.M.-T.). Inclusion criteria are (1) a single LH ischemic stroke, (2) a diagnosis of aphasia according to the Montreal–Toulouse aphasia battery [40], (3) the presence of anomia according to a standardized naming task [41], or complaint of anomia in everyday life, and (4) being right-handed prior to the stroke [42]. Exclusion criteria are (1) the presence of a neurological or psychiatric diagnosis other than stroke, (2) incompatibility with fMRI testing, or (3) diagnosis of mild cognitive impairment or dementia before stroke [43]. Table 1 contains sociodemographic information of the included participants, and Figure 1 shows their structural magnetic resonance imaging (MRI) results. All participants are francophones and live in the province of Quebec.

### 2.2. Experimental Procedure

The experimental protocol is similar to previous studies conducted in our lab [34,44,45,46,47]. A baseline evaluation was completed prior to therapy, including a language assessment and an initial fMRI session (T0). Following the baseline evaluation, participants received 3 one-hour therapy sessions over five weeks from a trained SLP (M.M.-T.). A second language evaluation and fMRI session (T1) were performed within a week of the end of therapy. This allowed us to identify therapy-induced language changes and neuroplasticity in rsFC.

#### 2.2.1. Language Assessment

Participants completed a comprehensive battery of language tests before (T0) and after (T1) Fr-PCA therapy (including the *Test de dénomination de Québec-60* (TDQ60) [41] for picture-naming; the *Test de dénomination de verbes lexicaux en images-38* (DVL38) [48] for verb naming, oral comprehension, repetition, and verbal fluency with semantic criteria sub-tasks from the Beta-86 *Montreal–Toulouse* aphasia battery [40]; and a narrative discourse sample of the Cinderella story [49,50].

The tests were administered by an SLP and split into three balanced 2.5-h assessment sessions, allowing us to collect three picture-naming baselines. The picture-naming baseline task comprised 270 pictures selected from the Bank of Standardized Stimuli BOSS [51] and previously validated among healthy elderly French speakers [52], including various categories such as fruits, vegetables, clothes, animals, body parts, furniture, and other objects. The baseline picture-naming task, repeated in all three assessment sessions, allows generating two independent lists balanced for frequency, number of phonemes, and syllables [53]. List 1 was composed of 20 items chosen with the participant to be trained in therapy, and list 2 was composed of 40 untrained items (i.e., to measure generalization). Items selected for lists 1 and 2 were named incorrectly (or with a response time greater than 10 s) on 2 or 3 baselines.

#### 2.2.2. Fr-PCA Therapy

Therapy followed the protocol developed by Leonard et al. [20]; it incorporated cues based on the phonological components of target words. The phonological components trained were the first sound of the target (question: “What sound does it start with?”), the final sound (“What sound does it end with?”), the number of syllables (“How many beats does the word have?”), a first sound associate (“What other word starts with the same sound?”), and a rhyme to the target (“What does this rhyme with?”). The PWA was asked to generate the answers to each cue regardless of if the target word is initially found or not; if this was not possible, they could choose between three valid options to promote active participation [54]. Participants received Fr-PCA using the items on list 1 (see above), for a total of 15 one-hour sessions, with a frequency of three sessions per week for a total of five weeks starting immediately after the assessment. According to the participant’s pace and tolerance, all items on list 1 were repeated 1 to 4 times during the session. Fr-PCA was presented on a laptop and coded into a python program, allowing for automatic randomization of items from session to session and online scoring for correct answers. The SLP controlled the therapy display on the laptop and gave appropriate feedback during the therapy.

Fr-PCA follows principles of neuroplasticity induced by experience introduced by Kleim and Jones [55] and adapted to treatment of stroke-induced aphasia by Kiran and Thompson [8], such as specificity of the strategy used (phonological cue generation), salience (by involving the participants in choosing the words to be trained), repetition, and intensity, promoting generalization and using complexity to promote learning.

#### 2.2.3. Outcome Measures

The primary outcome measure is the accuracy of confrontation naming on the therapy list (list 1). Picture-naming was recorded pre- and post-therapy (T0 and T1) without any cues nor feedback. Correct answers were scored as 1, and incorrect answers or answer latency superior to 10 s [56] were scored as 0.

Secondary outcome measures were the accuracy of confrontation naming on the untrained list (list 2) and standardized language tests.

### 2.3. Data Acquisition and Preprocessing

#### 2.3.1. Functional Neuroimaging Parameters

Images were acquired using a 3 T MRI Siemens Trio scanner updated to Prisma Fit with a standard 32-channel head coil during data collection. The image sequence is a T2*-weighted pulse sequence (184 volumes; TR = 2200 ms; TE = 30 ms; matrix = 64 × 64 voxels; FOV = 210 mm; flip angle = 90°; slice thickness = 3 mm; acquisition = 36 slides in the axial plane with a distance factor of 25% to scan the whole brain). A high-resolution structural image was obtained using a 3D T1-weighted imaging sequence using an MP-RAGE (TFE) sequence (TR = 2300 ms; TE = 2.98 ms; 192 slices; matrix = 256 × 256; voxel size = 1 × 1 × 1 mm^3^; FOV = 256 mm).

#### 2.3.2. Resting-State Acquisitions

During the resting-state acquisition that lasted 6:44 min, participants laid supine on the MRI scanner bed with their head stabilized. A black screen with a white point was presented. The participants were instructed to look at this point, relax and remain still and awake.

#### 2.3.3. Preprocessing

Data preprocessing was performed using SPM12 (Wellcome Trust Centre for Neuroimaging) implemented in MATLAB (version R2016a, MathWorks, Natick, MA, USA). The first five volumes were discarded as dummy scans. Images were slice-time corrected with reference to the acquisition time of the middle slice and motion-corrected with 12 motion parameters realignment to the first volume. Outliers were detected using ART (available online: http://www.nitrc.org/projects/artifact_detect, accessed on 6 September 2016) and defined as volumes with realignment parameters >2 mm and 2 degrees, or with signal intensity changes >4 times the standard deviations, parameters suggested as liberal settings [57] and used in several recent studies [34,58,59,60]. The T1 structural volume was co-registered and segmented into gray matter, white matter, and cerebrospinal fluid (CSF). Due to extensive lesions altering anatomical data processing, the spatial normalization into the MNI space was estimated from the functional data using an EPI template, with the SPM12 default values (affine regularization with ICBM/MNI space template; nonlinear frequency cut-off: 25; nonlinear iterations: 16; nonlinear regularization: 1). Images were resampled at 2 × 2 × 2 mm^3^ using a 4th-degree B-Spline interpolation. The anatomical data were spatially normalized using the same deformation parameters. The normalized volumes were visually checked with the implemented quality assurance tool, based on slice display with MNI boundaries, to prevent abnormal results (Appendix A). Since the signal of interest was considered to belong to the gray matter, the white matter, CSF, realignment, and scrubbing parameters were used as confounders for nuisance regression, with linear detrending. The lesions were segmented as CSF and regressed out. A band-pass filter (0.008–0.09 Hz) was applied.

### 2.4. Data Analysis

#### 2.4.1. Behavioral Responses to Therapy

Statistical analyses were performed using SPSS 26. The effect of Fr-PCA was measured using a Wilcoxon signed-rank test at the group level to compare the accuracy on the trained items (list 1) and the untrained items (list 2) at baseline (T0) and post-therapy (T1). The Wilcoxon signed-rank test is preferred, as this non-parametrical test is appropriate for a repeated measure design. Effect size was calculated with r = Z/√N [61] where ∼0.1 is considered a small effect, ∼0.3 an average effect, and ∼0.5 a large effect [62].

#### 2.4.2. Functional Connectivity Analysis

Resting-state functional connectivity analysis was performed using the CONN functional connectivity toolbox (v.18.b, available online: http://www.nitrc.org/projects/conn, accessed on 29 January 2021, [63]). An ROI-to-ROI analysis was conducted by computing the Fisher-transformed bivariate correlation coefficients between the time-series of each pair of ROIs. The bivariate correlation is preferred to facilitate comparing results throughout studies in the field in the future [34,60]. The whole brain was parcellated into 132 structurally homogenous ROIs, per the FSL Harvard-Oxford atlas for the gray matter and subcortical regions. In total, 106 ROIs were selected in the CONN atlas (91 cortical and 15 subcortical ROIs from Harvard-Oxford Atlas) and used to conduct an ROI-to-ROI analysis, creating a 106 × 106 FC matrix. ROIs from cortical and subcortical atlases were included in the analysis, as core areas in the language processing network are located in these regions. Without minimizing the involvement of the cerebellum in language processing, and more specifically phonological processing, the methodological decision to focus the analysis on cortical and subcortical areas was made. Functional connectivity was assessed, considering significant connection values for *p*-FDR corrected < 0.05. In the second-level analysis performed in CONN, a paired *t*-test was completed between pre-therapy and post-therapy connectivity matrices (post > pre, *p*-FDR corrected < 0.05). To assess the effect of initial aphasia severity on therapy-induced rsFC changes, a linear regression analysis was conducted with the initial BDAE aphasia severity score as regressor, where 1 is very severe aphasia, and 5 is mild aphasia.

Furthermore, to investigate the link between rsFC changes and naming improvements after therapy, we applied Spearman’s correlation analyses between ROI-to-ROI (R2R) rsFC changes (found significant in the main analyses), and the improvements on trained and untrained items.

## 3. Results

### 3.1. Behavioral Results

At the group level, Fr-PCA led to significant improvement on trained items (W(10) = 55.0, Z = 2.805, *p* = 0.005, r = 0.89) and untrained items (W(10) = 55.0, Z = 2.807, *p* = 0.005, r = 0.89). As shown in Figure 2, the mean improvement was 42% ± 25 for the trained items and 19% ± 15 for the untrained items. Table A1 in the Appendix B shows the performance for each participant on the language tests.

### 3.2. Functional Connectivity Results

#### Therapy-Induced rsFC Changes

Following Fr-PCA therapy, significant changes in rsFC were observed; statistical values are shown in Table 2. Namely, the anterior division of the left temporal fusiform cortex was more functionally connected to the supracalcarine (SCC) cortex bilaterally. Furthermore, the left SCC cortex was more functionally connected with the anterior division of the left inferior temporal gyrus. Finally, the lingual gyrus was less functionally connected with the superior frontal gyrus in the RH.

An effect of the initial aphasia severity on Fr-PCA therapy-induced rsFC changes was observed between the posterior division of the left temporal fusiform cortex and both the anterior division of the right superior temporal gyrus and the right insular cortex: the milder the aphasia, the greater the increase in functional connectivity between these regions. There was also an effect on the functional connectivity between the right frontal operculum cortex and the right pallidum. Statistical values are shown in Table 3.

### 3.3. Correlations between rsFC Changes and Naming Improvements following Therapy

Spearman’s correlation was performed between individual beta values for the significant rsFC changes identified and the improvements for trained and untrained items. Within the group of 10 participants with aphasia, no significant correlation was found between the rsFC changes and the main language outcome measures.

## 4. Discussion

The purpose of this study was to investigate therapy-induced naming improvement following Fr-PCA in persons with chronic aphasia while examining rsFC modifications. This is the first study to measure the efficacy of PCA in French, together with related rsFC changes. Ten participants with aphasia participated in this study; all of them were in the chronic phase and presented post-stroke aphasia characterized by anomia.

In line with previous literature on PCA in English [19,20,22,25,26,27,28,29,30], the results of this study show a significant improvement on trained items and a generalization of therapy effects to untrained items in French. Changes were also observed at the neurofunctional level. More specifically, there was an increased rsFC measured between ROIs known to support language processing and visual processing. Furthermore, a decrease in FC was observed between ROIs in the RH, particularly in the visual network and homolog areas to the language network. As participants were in the chronic stage of aphasia, spontaneous cerebral reorganization was unexpected. It is to be noted that there was no statistically significant correlation between the rsFC changes and behavioral improvements on the trained and untrained items. Nonetheless, these results support using rsFC to measure therapy-induced neuroplasticity in participants with aphasia and are discussed in detail below.

All the participants in the present study benefitted from Fr-PCA, a specific therapy protocol targeting naming recovery in aphasia by explicitly training the generation of phonological cues using picture stimuli and written supports. The goal following Fr-PCA is to strengthen the links between the semantic concepts and their phonological representations and enable the persons with aphasia to self-cue themselves. Previous studies have also shown that phonological cues are more beneficial to improve naming in persons with aphasia presenting variable anomia breakdown locus, i.e., showing either more semantic or phonological errors, when compared to semantic cues [28,64]. For the ten participants in this study, it could contribute to explain why, even if they presented variable anomia breakdown loci, they all benefited from FR-PCA. Moreover, as per previous work in our lab [34,44,45,46,47] and following the original protocol [20], frequency and dosage (3 times per week for 5 weeks total, approximately 60 min sessions) ensure the feasibility of Fr-PCA in the clinical practice in Quebec. Moreover, these therapy variables correspond to optimal variable ranges in terms of minutes per session [65,66], the number of sessions weekly [65,66], and cumulative dosage [66] reported in previous studies on naming therapies that improve naming performance in persons with aphasia.

Furthermore, in the Fr-PCA therapy protocol, when participants were unable to generate the cues themselves, three valid options were given to them so they can choose one, which contributed to active participation and engagement in the therapy process, two factors previously demonstrated to promote recovery effectively [67]. Another interesting feature of the Fr-PCA is the meaningfulness of words included in the training list for the person with aphasia. Indeed, among the words incorrectly named pre-therapy, the person chooses the most significant ones to be included in the therapy. This is in line with the principles of neuroplasticity adapted to the treatment of stroke-induced aphasia [8,55], which encourage clinicians to use salient therapy material. The results of this study indicate that the Fr-PCA protocol is a reproducible therapy protocol and opens a window into the study of mechanisms of neuroplasticity induced by Fr-PCA.

As mentioned previously, to the best of our knowledge, only three previous studies report neuroimaging data following PCA therapy [20,27,36]. Regarding the FC changes found in Marcotte et al. [22] and Rochon et al. [29], no common changes were found in our study. However, this could be related to the fact that the perspective is different (segregative versus integrative), making it difficult to reconcile the different results obtained. Furthermore, our results differ from van Hees et al. [38]. This may be due to the method used. Van Hees et al. [38] performed an ALFF analysis within a mask of LH language-related regions and their RH homologs. In contrast, analyses in the present study were performed with a whole-brain ROI-to-ROI method, allowing us to identify rsFC changes within and beyond the well-known language network.

Specifically, following Fr-PCA, functional connectivity changes between ROIs in the visual and language processing networks were observed. Increased rsFC was found between the anterior division of the left temporal fusiform cortex (BA20) and the SCC (BA17) bilaterally. Moreover, there was increased connectivity between the left SCC and the anterior division of the left ITG (BA20). BA20 has been previously linked to lexico-semantic processing for written words [68], to language production [69], and described as a potential language association area by Ardila et al. [70]. The written phonological cues, as well as the use of pictures in the Fr-PCA, may have contributed to the increased FC between these networks.

Enhanced functional connectivity in the LH was expected following therapy. Thus, it is well accepted that undamaged LH ROIs are key components of the network engaged in language recovery [7]. Although Fr-PCA promotes phonological strategies, no changes in functional connectivity between ROIs in the dorsal stream of the language network were observed. This observation is in line with a recent meta-analysis on treatment-related brain changes in aphasia; Schevenels et al. [71] discuss how phonological therapies induce neurofunctional changes beyond regions typically associated with phonological processing such as the SCC. Moreover, the location of treatment-related changes does not clearly depend on the type of language processing targeted [71]. Furthermore, it has previously been pointed out that picture stimuli are used in the PCA protocol, thus incorporating a semantic component [20]. Therefore, there is a possibility that this semantic factor, along with the fact that the protocol requires that all generated cues be written on the therapy board, may have induced increased connectivity between semantic and visual networks, allowing them to work in a more efficient matter.

Furthermore, going in the same direction as theories regarding maladaptive effects of RH recruitment on language function following aphasia [8,14,15,16], the results of this work show that naming recovery following Fr-PCA happened concomitantly with a decreased RH functional connectivity between visual (lingual gyrus, BA18) and semantic (SFG, BA8–9) networks [7]. Given the role of these ROIs in phonological processing, this finding might reflect the contribution of Fr-PCA in changing the language processing networks [7], with enhanced connectivity shifting back to the LH, concurrently with behavioral improvements measured. Specifically, the lingual gyrus—known as the secondary visual cortex—is activated following PCA, and particularly during a phonological judgment task [29], while the SFG supports phonological processing [72], and is considered a domain-general area recruited in support to the damaged network [7].

Interestingly, when looking at the effect of the initial severity of aphasia on the changes in rsFC following Fr-PCA, interhemispheric and RH increases in rsFC are observed with milder pre-therapy aphasia. In line with previous research, it could have been expected that milder aphasia would have recruited more ROIs in the spared LH [73]. However, more recently, Kiran and Thompson [8] identified an extended language processing network containing bilateral regions included in the traditional language network and domain-general areas. Hence, the increased connectivity found in milder aphasia between the spared LH language network (posterior division of the left temporal fusiform cortex, BA20) and contralateral ROIs involved in language processing (anterior division of the right STG (BA22) and the salience network (right insular cortex, BA13) is in line with previous works showing the role of the insula in phonological therapy efficacy [74]. Further studies could compare these results with healthy elderly controls, as this could represent a normalization of a more distributed language network. Increased functional connectivity was also observed between the right frontal operculum (BA45)—associated with speech, intonation, and music production [75]—and the right pallidum. Furthermore, the right frontal operculum sustains proper performance of multiple receptive and expressive auditory abilities, which are also put into play in the context of Fr-PCA [74]. Finally, these findings are in line with a recent meta-analysis on treatment-related brain changes in aphasia, showing that language recovery is not only associated with traditional language-related areas (in the left and right hemisphere) but also with more medial and subcortical areas [71]. In particular, the basal ganglia have also been reported to sustain recovery following phonological therapy [71]. Although the investigation of the potential role of lesion load on the post-therapy functional connectivity changes is beyond the scope of this paper, it is likely that lesion load can potentially account for some of the effects of aphasia severity on the changes in functional connectivity.

It is important to recognize that no correlation between the changes in rsFC and the improvements on naming trained and untrained items emerged from the results. Upon observing the data more closely, the lack of significant correlation seems to emerge from variable aphasia profiles, despite an effort to have a homogeneous group. Still, considering that over 5 weeks, healthy elderly controls demonstrated stable rsFC [34] and that in chronic persons with aphasia, without interventions, brain activity remains stable [76], it is reasonable to believe that the changes observed in these results could be the result of the Fr-PCA intervention. Further research is needed, however, to confirm this presumption.

This is the first study investigating functional connectivity changes following Fr-PCA using a set of validated stimuli in French, and applying Fr-PCA in participants with chronic aphasia. Although a group of 10 participants could be considered as a small, it represents the biggest number of participants compared to previous studies investigating the effect of PCA [19,20,22,30] (equal to [23]) and more or equal than 25 out of the 32 studies included in the meta-analysis studying therapy-induced brain changes [71]. Moreover, the absence of a control group is acknowledged by the authors. In a previous publication, it was shown with a group of healthy elderly controls that after a period of 5 weeks, there were no spontaneous rsFC changes [34]. However, considering that the reported changes are concomitant with positive therapy effects, we are confident that these changes reflect Fr-PCA induced neuroplasticity in rsFC because all ten participants were in the chronic phase, receiving no other therapy, language or other, than Fr-PCA and none of them changed any of their habits during the study. Indeed, it has been previously shown that persons with chronic aphasia show stable cortical activity in repeated fMRI sessions without therapy, which was not the case in this study [76]. Future studies will provide further indications for a better understanding of rsFC changes induced by specific therapy aiming to improve language and communication in aphasia. Semi partial correlations should be investigated in further studies with larger groups.

Employing rsFC within a clinical population such as persons with aphasia is not without its challenges. In this paper, the authors wish to disclose two limitations from the rs-fMRI data preprocessing. First, regarding the outlier detection, although the parameters used are commonly applied in the clinical studies [34,58,59,60], using liberal settings leads to higher frame-wise displacement values, which can have an impact on the results. Second, the lesion was segmented and regressed out as CSF. Previous studies have outlined that additional preprocessing steps might be necessary to remove potential artifacts from the lesion during rsFC analysis [77]. In this case, however, the impact on the results seems minimal as the areas involved in significant changes in rsFC are far from the lesion location. Furthermore, as this study is a pre-, post-therapy design with participants in the chronic stage, it is fair to believe that the signal would not change at both acquisition times within the necrotic tissue. Thus, because the participants are compared with themselves, this would not vary and limit the effect on analysis.

## 5. Conclusion

The present study shows that Fr-PCA leads to naming improvements with treated and untreated items, in participants with a variety of chronic aphasia profiles, and sizeable lesions. The results thus replicate previous research with PCA in English [19,20,22,23,30]. These improvements cooccurred with neurofunctional changes in rsFC, particularly between language and visual networks areas, thus suggesting that the nature of Fr-PCA and the strategies implemented during therapy modulate neuroplasticity as represented by functional connectivity changes. Hence, the present results contribute to provide evidence of therapy induced neuroplasticity resulting from specific therapy approaches integrating principles of experience-dependent neuroplasticity such as intensity and repetition, specificity of the strategy and salience of the stimuli [7]. A novel contribution to the study of language recovery in stroke-induced aphasia in the present study is that for the first time the therapy induced neurofunctional changes following Fr-PCA are approached within a whole-brain rsFC design which allows for the identification of functional changes outside the canonical language network. Considering the diversity of strategies integrated in different aphasia therapy protocols, the whole brain resting state connectivity approach supported by the present study opens a new window onto a more comprehensive understanding of the impact of language therapy on brain connectivity and behavior following post-stroke aphasia.

## Figures and Tables

**Figure 1 brainsci-11-01458-f001:**
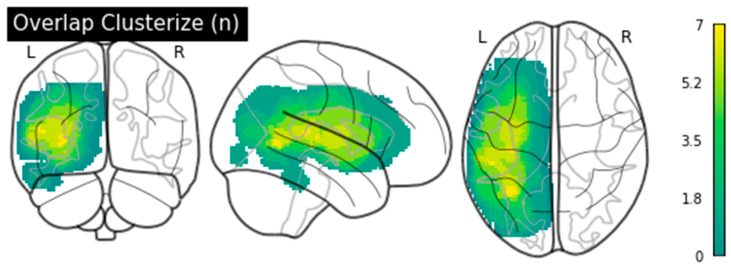
Participants’ lesion distribution in a glass brain. Color coding reflects the number of patients (1–7) with lesion overlap.

**Figure 2 brainsci-11-01458-f002:**
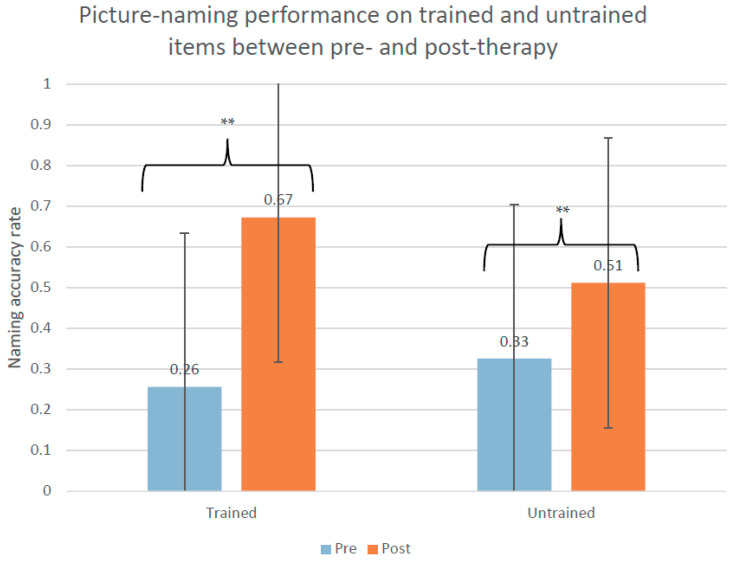
Picture-naming performance on trained and untrained items pre-therapy and post-therapy. ** *p* = 0.005.

**Table 1 brainsci-11-01458-t001:** Sociodemographic and clinical data for the participants.

ID	Sex	Age	Years of Education	Time Post-Onset (Months)	Lesion Size (mm^3^)	Aphasia Type	Aphasia Severity (BDAE Scale)	% Noun Naming (TDQ60)
PCA1	M	73	8	36	3188	Transcortical motor	4	0.40
PCA2	M	82	15	24	138,096	Transcortical mixed	2	0.27
PCA3	M	48	15	22	26,833	Transcortical motor	3	0.72
PCA4	W	70	15	41	124,217	Global	1	0.02
PCA5	M	60	12	172	223,253	Anomic	4	1.00
PCA6	W	72	12	47	95,672	Broca	2	0.30
PCA7	M	65	15	57	104,924	Anomic	2	0.95
PCA8	W	63	18	11	66,573	Broca	2	0.95
PCA9	M	79	20	12	43,121	Global	1	0.15
PCA10	M	77	17	11	12,874	Anomic	3	0.60

**Table 2 brainsci-11-01458-t002:** Functional connectivity changes following Fr-PCA therapy.

Region A	Region B	T(9)	*p*-FDR
ant. Temporal Fusiform Cortex L	Supracalcarine Cortex L	7.20	0.0053
	Supracalcarine Cortex R	4.83	0.0488
Supracalcarine Cortex L	ant. Inferior Temporal Gyrus L	5.07	0.0443
Lingual Gyrus R	Superior Frontal Gyrus R	−5.73	0.0298

**Table 3 brainsci-11-01458-t003:** Functional connectivity changes following Fr-PCA therapy with initial aphasia severity as a regressor.

Region A	Region B	T(8)	*p*-FDR	R^2^
post. Temporal Fusiform Cortex L	ant. Superior temporal gyrus R	10.82	0.0005	0.94
	Insular cortex R	5.23	0.0413	0.77
Frontal operculum cortex R	Pallidum R	9.24	0.0016	0.91

## Data Availability

The data presented in this study are available on request from the corresponding author. The data are not publicly available due to legal issues.

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
