# Peer review of "Resting-State Functional Connectivity following Phonological Component Analysis: The Combined Action of Phonology and Visual Orthographic Cues"

_brainsci, 2021, doi:10.3390/brainsci11111458_

Round 1
Reviewer 1 Report
This paper indicates the positive impact of PCA treatment to the subjects with chronic aphasia. Improved behavioral performance, and altered functional connectivity have been reported. But I do have some concerns:
(1) fMRI processing:
(a) Realignment parameters are regressed out. However, this is not clear to us how many motion parameters are included. 6 motion parameters, 12 motion parameters, or 24 motion parameters? Please clarify.
(b) And please also provide the mean frame-wise displacement value of each subject. Subjects with excessive mean FD value cannot be included in the analysis.
(c) Using EPI image to perform spatial normalization is inaccurate. I understand this is a trade-off due to the influence of lesions. To convince us that the registration quality is acceptable, please provide the registration quality comparison results of each subject, by overlapping co-registered image to EPI template. A code from FSL could provide this image: slicer co-register.nii.gz EPI_template.nii.gz -a test.png
(2) While to exclude the 58 regions from the atlas with 164 ROIs?
(3) Could the alternation of FC correlates with the alternation of behavioral performance? If so, the finding is much more attracting.
(4) The variation of time post-onsite and lesion size is very large. Do they have any effect to the statistical results?
(5) The detected regions are mostly associated visual function. The major language regions: Broca and Wernicle regions are not detected. Could this due to the fact that PCA treatment only has a major impact on visual recognition?
Reviewer 2 Report
The study examines changes in resting state functional connectivity in ten patients with chronic aphasia receiving phonological naming therapy. The goal is important, however, the lack of a control group and the absence of any correlation between behavioral treatment outcomes and connectivity changes make it hard to attribute the changes in connectivity to the treatment. Furthermore, there are a number of unclear methodological choices and unjustified conclusion.
- How do the authors deal with potential artifacts from the lesion area that could confound resting state connectivity measures? (See: Yourganov G, Fridriksson J, Stark B, Rorden C. Removal of artifacts from resting-state fMRI data in stroke. NeuroImage: Clinical. 2018;17:297-305). If they do not, what may have been the effects of this on the findings?
- What is the reason for using bivariate correlation coefficients in CONN rather than semi partial correlations, that control for the multi-collinearity among ROIs, and enable more specific identification of the pairs of connected ROIs?
- What was the rationale for selecting these 106 ROIs out of the entire 164 of the whole brain? Why not more or less or others?
- While the authors hypothesize that “the underlying processes for therapy will influence the rsFC” (line 87) to test this hypothesis in the absence of a control group they should test the correlation between behavioral effects of treatment and connectivity changes. Such testing was not reported.
- The conclusion that “recovery following Fr-PCA was associated with a decreased RH functional connectivity” (line 350) is not justified, as no correlation was shown between recovery and connectivity changes. Moreover, in the absence of such correlation it is impossible to conclude that this reduction in RH connectivity, is in line with the maladaptive effects of the RH on recovery.
- While the authors acknowledge the problem of not including a control group they downplay its effect by saying that all participants have benefited from therapy and did not undergo any other major changes during this period. Nevertheless, these points are not relevant as they cannot rule out the possibility that random changes in connectivity occur even without treatment.
More minor comments:
Line 45: the sentence talks about the network associated with “word naming” but cites models and studies (references 9-11) related to speech perception.
Line 176: how long was the resting state scan?
Line 218: it should be specified how the second level analysis was done, whether in CONN, or in another statistical software.
Line 226: it is unclear what is the “r” stand for (r=0.89)
Lines 259-261: what is the basis for referring to SFG as “language network”?
Lines 292-326: The paragraph is very hard to follow due to long lists of author names and anatomical labels. More integration is needed between studies to draw any conclusions or make comparisons with the current results.
Figure 1: The middle panel shows the right rather than the left hemisphere. The color scale should have a legend.
Figure 2: ‘treated’ and ‘trained’ are used inconsistently
Unclear or ungrammatical sentences or words in the following lines: Sentence on lines 54-56; Line 92: ‘objectified’; Line 127: ‘constituted’; Line 238: ‘correlated’ should be ‘connected’; Sentence on lines 272-274; Line 368: ‘controlateral’;
Round 2
Reviewer 1 Report
The authors have improved the manuscript. However, as shown in the cover letter, some subjects still have relatively large frame-wise displacement (FD), even after scrubbing process. This would influence the FC values. A common FD threshold for strict rsfMRI analysis is 0.5 mm. For this study, this needs to be considered as a limitation.
Reviewer 2 Report
The authors have addressed most of my concerns.
However, for points 1,2,3 - the rationale for using these specific methods, should also be explained in the text.
